# Energy Guided Geometric Flow Matching

## Abstract

A useful inductive bias for temporal data is that trajectories should stay close to the data manifold. Traditional flow matching relies on straight conditional paths, and flow matching methods which learn geodesics rely on RBF kernels or nearest neighbor graphs that suffer from the curse of dimensionality. We propose to use score matching and annealed energy distillation to learn a metric tensor that faithfully captures the underlying data geometry and informs more accurate flows. We demonstrate the efficacy of this strategy on synthetic manifolds with analytic geodesics, and interpolation of cell trajectories from single-cell RNA sequencing data.

## 1 Introduction

Generative models are the workhorse of modern AI, enabling us to sample from complex data distributions. Methods based on diffusion have recently excelled at mapping from Gaussian noise distributions to data manifolds, and flow models enable mapping from any noise distribution to the manifold. Typically, these methods are interested in the trajectory that an individual sample takes, from noise to data, only because it enables straightforward simulation with an ODE or SDE solver (Yang et al., 2024). However, for temporal data one may be interested in these trajectories themselves, for example in the context of single-cell trajectory inference.

Recently, more focus has been given to parameterize the trajectories along a manifold, e.g. with fitting expressive Neural ODEs. Early methods focused on the simulation-based setup that required differentiation through an ODE Solver (Tong et al., 2020). However, the improved efficiency of training simulation-free methods such as flow matching (Lipman et al., 2022; Tong et al., 2023) has granted simpler training to these methods.

We are especially interested in the application of inferring cell trajectories in single-cell genomics. Currently, many datasets are collected at a single timepoint, and temporal models must resort to "pseudotime" methods to artificially create multiple timepoints, but a growing number of datasets have subsets of cells measured at different times. In such settings, accurate recovery of manifold geometry is critical for characterizing cellular development and disease processes.

Our contributions are three-fold: (i) a novel combination of score-matching and annealed energy distillation to parameterize a metric tensor that characterizes the underlying data manifold; (ii) a variant of stratified sampling to robustly infer geometry from density, mitigating failures on disconnected components; and (iii) practical application of these metrics to synthetic manifolds with known geodesics and single-cell RNA trajectory inference.

## 2 Setup

### 2.1 Denoising Score Matching

The score matching objective (Song et al., 2020) is the backbone of modern diffusion models, providing an efficient way to learn a family of score functions corresponding to the data distribution under different levels of noise. For a full diffusion model parameterizing density over time, the model takes as input a time variable, but in our setting we consider a monotonic noise schedule of

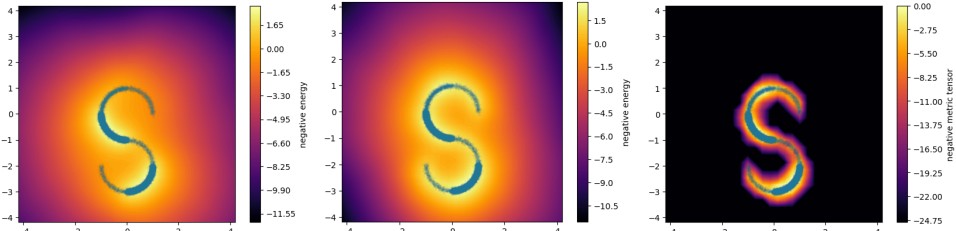

Figure 1: A visualization of our method. We fit an initial score and energy (left), then use annealing and self-normalized importance sampling to fit an updated energy that is less biased by unequal density in the data (middle), and we clip the energy to calculate a balanced metric tensor that captures the manifold (right).

$\sigma_{min} < \cdots < \sigma_{max}$ and simply condition on noise. In this case, the score matching loss is given by

$$\mathcal{L}_{score}(\theta) = \mathbb{E}_{x \sim p_{\text{data}}, \epsilon \sim \mathcal{N}(0,I)} \left[ \sum_i \left\| s_\theta(x + \epsilon \sigma_i, \sigma_i) + \frac{\epsilon}{\sigma_i} \right\|^2 \right]$$

which yields $s_\theta(x, \sigma_i) \approx \nabla \log p_{\sigma_i}(x)$ where $p_{\sigma_i}(x)$ is the density of $p_{data}$ convolved with Gaussian density of standard deviation $\sigma_i$.

There are many equivalent formulations of this loss that are more practical for optimization, where instead of learning $s_\theta$ directly one can instead learn to predict the noise via the loss function

$$\hat{\mathcal{L}}_{score}(\theta) = \mathbb{E}_{x \sim p_{data}, \epsilon \sim \mathcal{N}(0,I)} \left[ \sum_i \| \epsilon_\theta(x + \sigma_i \epsilon, \sigma_i) - \epsilon \| \right]^2$$

and approximate the score by $\nabla \log p_{\sigma_i}(x) \approx -\frac{\epsilon_\theta(x)}{\sigma_i}$. We will rely on this loss in all of our subsequent experiments.

## 2.2 CONDITIONAL FLOW MATCHING

Given two measures, conditional flow matching (Tong et al., 2023; Lipman et al., 2022) learns a vector field to map one distribution $p_0$ to another distribution $p_1$. A general form of this loss is given in Albergo et al. (2023) where one considers a coupling of $\pi$ of the two measures and a parameterization of paths $x_t := x_t(x_0, x_1)$ that defines a conditional density $p_t(\cdot|x_0, x_1)$. Then optimizing the loss

$$L_{flow}(\vartheta) = \mathbb{E}_{t \sim U([0,1]), (x_0, x_1) \sim \pi} \left\| v_\vartheta(x_t) - \frac{d}{dt} x_t \right\|^2$$

guarantees that $v_\vartheta$ solves the continuity equation with the marginal density $p_t$. In this way, one can numerically integrate $v_\vartheta$ to sample along trajectories from $p_0$ to $p_1$. When the goal is sampling, a common choice is $x_t = (1 - t)x_0 + tx_1 + \sigma_{flow}\epsilon$ where $\epsilon \sim \mathcal{N}(0, I)$ such that $p_t(x_0, x_1)$ is a Gaussian centered along the straight line between $x_0$ and $x_1$.

## 2.3 STOCHASTIC INTERPOLANTS ALONG GEODESICS

When a prior is available, flow matching trajectories may be intentionally biased towards a known or inferred manifold. Namely, given a metric tensor $\mathbb{G}(x)$, one may parameterize conditional paths between $x_0$ and $x_1$ of the form,

$$x_t := x_t(x_0, x_1) = (1 - t)x_0 + tx_1 + t(1 - t)\psi(x_0, x_1, t) + \sigma_{flow}\epsilon,$$

which are guaranteed to be geodesics after minimizing the loss,

$$\mathcal{L}_{geodesic}(\psi) = \mathbb{E}_{t \sim [0,1], (x_0, x_1) \sim \pi} \| \dot{x}_t \|_{\mathbb{G}(x_t)}^2 .$$

This characterization was first introduced in Kapusniak et al. (2024). Given satisfactory geodesics, one can make this choice of conditional paths and use the same conditional flow matching loss as usual.

## 3 RELATED WORK

### 3.1 FLOW MATCHING

Flow matching has rapidly evolved since its original formulations (Lipman et al., 2022; Albergo et al., 2023; Liu et al., 2022). In particular, research has focused on incorporating prior knowledge into the the definition of the conditional flows, based on optimal transport (Tong et al., 2023; Pooladian et al., 2023), minimal curvature (Rohbeck et al., 2025), and user-defined potential functions (Neklyudov et al., 2023), among others. While these methods enrich interpolation flexibility, they rely on Euclidean path parameterization or external priors, rather than directly learning a data-driven metric tensor.

### 3.2 ENERGY DISTILLATION FROM SCORE MATCHING

Although the primary application of score matching continues to be diffusion models (Song et al., 2020), several methods have focused on using score matching as a tool to distill an accurate energy of the data. Thornton et al. (2025) suggest a specific parameterization of the energy used for sequential Monte Carlo sampling, and Akhound-Sadegh et al. (2025) infer a score and energy simultaneously for annealed sampling. These methods, however, fail to connect energy to manifold geometry or flow interpolation.

### 3.3 LEARNING GEODESICS ALONG DATA-DRIVEN MANIFOLDS

Flow matching was initially extended to trajectories on manifolds by Chen & Lipman (2023) with a focus on settings where geodesics could be exactly or approximately computed. This was extended further by Kapusniak et al. (2024) allowing for learning of geodesics by minimizing energy with respect to some inferred metric tensor.

Other methods consider interpolation in a latent space and map back to ambient space to learn a curved path (Palma et al., 2025). de Kruiff et al. (2024) also considered learned metric tensors in latent space through a pullback, although they require simulation through a Neural ODE solver to train.

The methods most related to the current work also consider how to infer a non-trivial metric tensor from the data manifold to derive plausible geodesics and train simulation-free flows. Kapusniak et al. (2024) learn a metric tensor based on the linear combination of RBF kernels proposed in Arvanitidis et al. (2020). Similarly, Sun et al. (2024) learn a pullback metric using the pairwise distances inferred from the embedding method PHATE (Moon et al., 2019).

Additionally, so-called Fermat distances (Groisman et al., 2022) that define a metric tensor through a density have been studied before in more limited contexts, primarily for 2d inference of single geodesics in Sorrenson et al. (2024) and image interpolation in Yu et al. (2025). In contrast, we learn an adaptive energy metric tensor with annealing in our setting, and apply it to the task of learning general geodesics.

## 4 METHODS

We propose a multi-stage training procedure that yields a geometrically faithful flow-matching trajectory aligned with the data manifold. First, given samples $x \sim p_{\text{data}}$, we estimate a density $p_\theta(x)$ by jointly learning the score $s_\theta(x) = \nabla_x \log p_\theta(x)$ and the energy $E_\zeta(x)$ via score- and energy-matching. This estimation is iteratively refined to better align $p_\theta$ with $p_{\text{data}}$. Next, we construct a metric tensor $\mathbb{G}$ from the learned energy, and jointly learn geodesics and the associated geodesic distance $d$ under $\mathbb{G}$. Finally, we compute an optimal transport coupling under the cost defined by $d$ and train a flow-matching vector field to realize the resulting displacement interpolation, yielding a flow that respects the learned manifold geometry.

## 4.1 Score and Energy Matching

We first aim to learn the shape of the manifold through score and energy matching in the form of the data generating density. Formally, denote the score matching network as $s_\theta(x, \sigma)$ where $\theta$ is the set of learnable parameters and $\sigma$ is the noise scale. Denote noised data at noise scale $\sigma_i$ as $y = x + \epsilon \sigma_i$. We perform denoising score matching by minimizing,

$$\mathcal{L}(\theta) = \mathbb{E}_{x \sim p_{\text{data}}, \epsilon \sim \mathcal{N}(0, I)} \left[ \sum_i \left\| s_\theta(y, \sigma_i) + \frac{\epsilon}{\sigma_i \cdot \beta} \right\|^2 \right],$$

where the score net $s_\theta(\cdot, \sigma)$ takes noised data from various noise levels and conditions on an embedding of the noise scales. $\beta$ is the a temperature scalar. The log density of the learned generating distribution is then given by $s_\theta(\cdot, \sigma) \approx \nabla_x \log p_\theta(x)$. We can then recover energy as,

$$E(x) \approx -\log p_\theta(x),$$

which can be learned by optimizing

$$\mathcal{L}(\zeta) = \mathbb{E}_{x \sim p_{\text{data}}, \epsilon \sim \mathcal{N}(0, I)} \left[ \sum_i \| \nabla E_\zeta(y, \sigma_i) + s_\theta(y, \sigma_i) \|^2 \right]$$

where $\zeta$ is the set of parameters from the energy network $E_\zeta(., \sigma)$ which conditions on the noise embedding and takes in noised data.

## 4.2 Iterative Density Refinement

The primary issue with simply using the score is imbalance in data density. To better align the learned density $p_\theta$ to the underlying geometry, we repeat score and energy matching multiple times while refining the learned density with the following operations to achieve stable and annealed density estimation.

**Self normalized importance sampling (SNIS).** One important component of these refinement steps is self normalized importance sampling. At refinement step $k$, the reweighting is,

$$\hat{p}_{\text{data}}^k(x) \propto \frac{p_{\text{data}}(x)}{\exp(-\beta_w \text{clip}(E_\zeta^k(x, \sigma_{min})))},$$

where $\beta_w$ is a hyperparameter and the clipping is based on energy quantiles. We use this reweighting over our data, which is closer to uniform, in subsequent steps.

**Density annealing.** Aside from reweighting, we also give the option to anneal the density based on the previous step density and temperature. Let $y = x + \epsilon \sigma_i$, at refinement step $k$, the density annealing loss is given by,

$$\mathcal{L}^k(\theta) = \mathbb{E}_{x \sim \hat{p}_{\text{data}}^k, \epsilon \sim \mathcal{N}(0, I)} \left[ \sum_i \alpha \left\| s_\theta^k(y, \sigma_i) + \frac{\epsilon}{\sigma_i \cdot \beta_k} \right\|^2 + (1 - \alpha) \left| s_\theta^k(y, \sigma_i) - \frac{\beta_k}{\beta_{k-1}} s_\theta^{k-1}(y, \sigma_i) \right| \right]$$

when $\alpha = 1$ this is equivalent to optimizing denoising score matching with current temperature $\beta_k$ at step $k$.

The iterative density refinement is possible due to the scalability and efficiency of score/energy matching models. Our final energy estimation is based on the refined density after $K$ steps.

**Stratified Sampling.** One shortcoming of score matching is learning the correct normalization of isolated components (Wenliang & Kanagawa, 2020). Intuitively, this presents a challenge for using the score to induce a metric tensor as even a very accurate score may assign unequal densities to identical but separated components. We consider a novel approach for addressing this issue by introducing clustering and stratified sampling (Owen, 2013).

Let $p^*$ denote the uniform distribution on the underlying data manifold. Since our ultimate goal is to learn a density that is close to $p^*$, and typical notions of distances between measures like

integral probability metrics are based on integrating against test functions, we equivalently want a low-variance estimator for $\mathbb{E}_{x \sim p^*}[g(x)]$ for any test function $g$.

Thus, we consider a form of Rao-Blackwellization by first clustering the underlying data into clusters $C_1, \ldots C_J$, doing score matching and annealing on each cluster component independently, before finally reweighting to learn a score over the entire dataset. Formally, this loss is

$$\mathcal{L}^k(\theta) = \mathbb{E}_{j \sim [J], x \sim \hat{p}^k_{\text{data}}(\cdot | C_j), \epsilon \sim \mathcal{N}(0, I)} \left[ \sum_i s^k(x + \epsilon \sigma_i, \sigma_i, j) + \frac{\epsilon}{\sigma_i} \right]$$

When fully trained, this loss yields a score function that can be conditioned on any individual cluster, with a respective energy that can be distilled per cluster as well. The utility of this clustering is that, because the score can be better approximated per local cluster, the SNIS estimator for a uniform distribution on that cluster is more accurate, and therefore the combined reweighting across all clusters will be closer to uniform. Explicitly, we use the normalization

$$\hat{p}^k_{\text{data}}(x, j) \propto \frac{p_{\text{data}}(x)}{\exp(-\beta_w \text{clip}(E^k_\zeta(x, \sigma_{min}, j)))},$$

$$\hat{p}^k_{\text{data}}(x) = \sum_{j=1}^J \frac{\hat{p}^k_{\text{data}}(x, j)}{|C_j|}.$$

We use Leiden clustering (Traag et al., 2019) to infer our clusters, as is common in single-cell literature.

### 4.3 METRIC TENSOR AND LEARNING GEODESIC

Having estimated the energy $E^K$, we next construct a metric tensor based on the learned energy. More specifically, the metric tensor is,

$$\mathbb{G}(x) = \gamma + \text{clip}(\lambda \exp(E^K(x)))$$

where $E^K(x)$ is the learned energy after $K$ refinement steps, and we clip according to quantiles of the energy taken as hyperparameters, as specified in the Appendix. Similar to Kapusniak et al. (2024), we penalize trajectories passing through off-manifold regions with low data density by writing trajectory as,

$$x_t | x_0, x_1 = (1 - t)x_0 + tx_1 + t(1 - t)\psi(x_0, x_1, t) + \sigma_{flow}\epsilon$$

where $\psi$ is a neural network that aims to approximate the geodesic given the metric tensor. We learn $\psi$ by energy minimization, i.e. minimizing

$$\mathcal{L}(\psi) = \mathbb{E}_{t \sim \mathcal{U}[0,1], (x_0, x_1) \sim \pi} \left[ \|\dot{x}_t\|^2_{\mathbb{G}(x_t)} \right]$$

where $\dot{x}_t$ is the time derivative of $x_t$.

### 4.4 DISTANCE LEARNING AND FLOW MATCHING

We use metric learning to obtain a distance that is consistent with the metric tensor defined. More specifically, we learn an isometric embedding $f : (\mathcal{M}, \mathbb{G}) \mapsto (\mathbb{R}^d, < ., . >_{\mathbb{R}^d})$ that maps defined geodesic into a straight-line, constant-speed segment in Euclidean space. To achieve this, we optimize

$$\mathcal{L}(f) = \mathbb{E}_{t \sim \mathcal{U}[0,1], (x_0, x_1) \sim \pi} \left[ \|\partial_t f(x_t)\|^2 - \|\dot{x}_t\|^2_{\mathbb{G}(x_t)} \right]$$

where $\pi$ is the optimal coupling with the learned distance $d(x_0, x_1) = \|f(x_0) - f(x_1)\|$. Finally, flow matching is used to parameterize paths that satisfy the continuity equation. Denoting the flow network as $v(t, x_t)$, we optimize

$$\mathcal{L}(v) = \mathbb{E}_{t \sim \mathcal{U}[0,1], (x_0, x_1) \sim \pi, \epsilon} \left[ \|v(t, x_t) - \dot{x}_t\|^2 \right]$$

## 4.5 THEORETICAL ANALYSIS

In this section we show that with our stratified sampling approach, the model learns a density that is a mixture among the clusters. Let $\{C_j\}_{j=1}^J$ be a disjoint partition of a data manifold $\mathcal{M}$. Our aim is to estimate $\mu = \mathbb{E}_{p*}[g(x)]$ for some testing function $g(x)$. Assume cluster $C_j$ has size $n_j$, the within-in cluster proposal distribution is given by

$$q_j(i) := \frac{r_i}{\sum_{k \in C_j} r_k}, \qquad i \in C_j$$

where $r_i = f(E(x_i)) > 0$ is a positive function of learned energy. Set $R_j = \sum_{k \in C_j} r_k$, if the target distribution is uniform on cluster rescaled by number of clusters, i.e.

$$p_j^* = \frac{1}{n_j} \cdot \frac{1}{J}, \quad \mu_j = \frac{1}{n_i} \sum_{i \in C_j} g(x_i)$$

The importance weight for $i \in C_j$ is

$$W_j(i) \;=\; \frac{p_j^*(i)}{q_j(i)} \;=\; \frac{R_j}{J\, r_i}$$

Given i.i.d. samples $I_1, \ldots, I_{m_j} \sim q_j$, the importance sampling estimator within cluster is:

$$\widehat{\mu}_j^{\text{SNIS}} \;=\; \frac{\sum_{t=1}^{m_j} W_j(I_t)\, g(x_{I_t})}{\sum_{t=1}^{m_j} W_j(I_t)}.$$

Across all clusters, our target and estimator becomes

$$p^* = \frac{1}{J} \sum_{j=1}^J \text{Unif}(C_j), \quad \widehat{\mu}^{\text{SNIS}} = \frac{1}{J} \sum_{j=1}^J \widehat{\mu}_j^{\text{SNIS}}$$

With this setup we can introduce the following theorem to show estimation convergence:

**Theorem 4.1.** *Assume for each cluster $C_j$, $r_i > 0$ for all $i \in C_j$ and the importance weights have finite second moment under per-cluster measure $q_j$, as number of samples $m_j$ in $C_j$ goes to infinity we have*

$$\widehat{\mu}^{\text{SNIS}_j} \xrightarrow{p} \mu_j, \quad \widehat{\mu}^{\text{SNIS}} \xrightarrow{p} \mu$$

*Additionally, if $\mathcal{L}(x; \theta)$ is the denoising score matching loss, the objective converges to $\mathbb{E}_{p^\star}[\mathcal{L}(x; \theta)]$*

*Proof.* Since $W_j(i) \;=\; \frac{p_j^*(i)}{q_j(i)}$ and $I_j \overset{\text{i.i.d.}}{\sim} q_j$, then

$$\mathbb{E}_{q_j}[W_j(i)g(x_i)] = \sum_{i \in C_j} q_j(i) \frac{p_j^*(i)}{q_j(i)} g(x_i)$$

$$= \sum_{i \in C_j} p_j^*(i)g(x_i) = \frac{1}{J}\mu_j$$

We also have

$$\mathbb{E}_{q_j}[W_j] = \sum_{i \in C_j} q_j(i) \frac{p_j^*(i)}{q_j(i)} = \frac{1}{J}$$

Hence under the law of large numbers, we have

$$\widehat{\mu}^{\text{SNIS}_j} = \frac{\sum_{t=1}^{m_j} W_j(I_t)\, g(x_{I_t})}{\sum_{t=1}^{m_j} W_j(I_t)} \xrightarrow{p} \frac{\mathbb{E}_{q_j}[W_j(I_t)g(x_i)]}{\mathbb{E}_{q_j}[W_j]} = \frac{(1/J)\mu_j}{(1/J)} = \mu_j$$

Since global estimator is given by averaging over clusters, we have

$$\widehat{\mu}^{\text{SNIS}} = \frac{1}{J} \sum_j \widehat{\mu}_j^{\text{SNIS}} \xrightarrow{p} \frac{1}{J} \sum_j \widehat{\mu}_j = \mu$$

Additionally, if we set $g(x_i) = \mathcal{L}(x_i; \theta)$ as the testing function, the SNIS estimator becomes a risk estimator

$$\widehat{\mathcal{R}}_j^{\text{SNIS}} = \frac{\sum_{t=1}^{m_j} W_j(I_t)\mathcal{L}(x_{I_t}; \theta)}{\sum_{t=1}^{m_j} W_j(I_t)}$$

Hence

$$\widehat{\mathcal{R}}_j^{\text{SNIS}} \xrightarrow{p} \frac{\mathbb{E}_{q_j}[W_j(I_t)\mathcal{L}(x_{I_t}; \theta)]}{\mathbb{E}_{q_j}[W_j]} = \frac{\sum_{i \in C_j} p_j^*(i)\mathcal{L}(x; \theta)}{\sum_{i \in C_j} p_j^*(i)} = \mathbb{E}_{p_j^*}[\mathcal{L}(x; \theta)]$$

Similar mapping from cluster specific estimator to global estimator holds, hence

$$\widehat{\mathcal{R}}^{\text{SNIS}} \xrightarrow{p} \mathbb{E}_{p^*}[\mathcal{L}(x; \theta)]$$

$\square$

## 5 RESULTS

### 5.1 BASELINES

We consider comparison against two primary methods, namely CFM where the metric tensor is constrained to be the identity (and therefore geodesics and a distance embedding don't need to be learned), and MFM where the metric tensor is given by a conformal metric defined by a linear combination of RBF kernels as in Arvanitidis et al. (2020). We consider GAGA (Sun et al., 2024) as another worthwhile baseline, but unfortunately we were unable to reproduce and run their publicly available code due to issues with CUDA memory and heldout test data used in the definition of their graph-based metric tensor.

### 5.2 SYNTHETIC DATA

We first consider settings where we can obtain analytic, closed form characterizations of the geodesics to measure against the learned geodesics. In those cases, we measure using an Average Geodesic Error calculated as

$$\mathcal{L}_{AVE}(\gamma, \gamma^*) = E_{t \sim U(0,1),(x_0,x_1) \sim \mu \times \mu} \|\gamma(x_0, x_1, t) - \gamma^*(x_0, x_1, t)\|^2$$

where $\gamma$ is the learned geodesic, $\gamma^*$ is the analytic geodesic, and $\mu$ is some base measure on the underlying manifold, for example the uniform Haar measure.

We consider data sampled uniformly on the surface of a sphere in varying dimensions. Note that while this task is handled by RFM (Chen & Lipman, 2023) if the manifold is known a priori, learning it directly from the data requires parameterizing the geodesics, and therefore provides a setting to check how accurately our learned geometry matches the ground truth. We omit comparison against CFM, as the optimal solution is simply straight geodesics that ignore the geometry of the sphere entirely.

Table 1: Comparison of average geodesic error along sphere geodesics in varying dimensions.

| Method | Dimension | AVE |
|--------|-----------|-----|
| MFM | 10 | $.21 \pm .07$ |
| EGGFM | 10 | $\mathbf{.12 \pm .07}$ |
| MFM | 20 | $.20 \pm .05$ |
| EGGFM | 20 | $\mathbf{.19 \pm .05}$ |

We observe that EGGFM outperforms MFM in this synthetic setting where we can measure correct geodesics exactly. Visually, we can observe 2d slices of the energy of EGGFM (which is equivalent to the metric tensor up to rescaling) and the metric tensor of MFM in Figure 2. Empirically the energy-induced metric is accurate in projected space, while the MFM metric is parameterized through RBF kernels around individual points. Although we found the best performance for MFM

using a total of 2000 points to parameterize the metric, the curse of dimensionality makes it difficult to cover even a moderately high dimensional manifold this way, and hence most of the points have no intersection with the 2d plane.

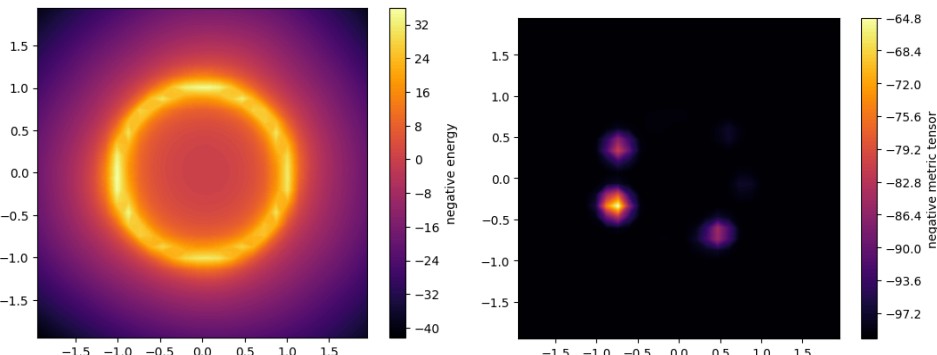

Figure 2: The negative energy of EGGFM (left) and negative metric tensor of MFM (right) on the 10 dimensional sphere, projected onto the first two dimensions.

### 5.3 INTERPOLATION OF SINGLE-CELL RNA DATA

We then consider the case of interpolating temporal single-cell RNA data. We hold out single time points for testing and measure model performance with one Wasserstein distance.

#### 5.3.1 EMBRYONIC BODY DATASET

We test our model on the Embryonic Body (EB) dataset introduced by Moon et al. (2019) and processed by Tong et al. (2020). The dataset consists of 5 time points over 30 days, which we denote with index 1-5. In our experiments, we hold out time points 2, 3, and 4 and train separate models on the full-time scale. We compare our model with conditional flow matching(CFM) and metric flow matching (MFM)(Kapusniak et al. (2024)). We compute distances in a PCA representation, a standard practice in single-cell RNA-seq analysis that reduces noise while extracting dominant biological variation.

Table 2: Wasserstein 1 distance averaged across left-out marginals($\downarrow$ better) for 5-dim PCA representation of the EB dataset. Results are averaged across 3 independent runs.

| Method | W1 distance($\downarrow$) |
|--------|---------------------------|
| CFM | $0.711 \pm 0.018$ |
| MFM | $0.727 \pm 0.042$ |
| EGGFM | $\mathbf{0.674 \pm 0.039}$ |

In Table 2, we observe significant performance improvement for EGGFM, which demonstrates the benefit of constraining learned trajectories to flow through the actual data manifold. In contrast, the weaker performance of MFM highlight the fact that even in relatively low-dimensional manifolds (5d), local RBF-kernel based estimation of the geodesic is insufficient to capture data geometry.

#### 5.3.2 HEMATOPOIETIC STEM CELL DATASET

We additionally apply our method to a low-dimensional projection of the CITE sequencing dataset (measuring RNA and protein features) of hematopoietic stem cells (Burkhardt et al., 2022). This dataset includes four timepoints on days 2, 3, 4, and 7, and therefore we can only evaluate on two heldout timepoints that have a distribution before and after. Our metrics in Table 3 show our method continues to show favorable results.

Table 3: Wasserstein 1 distance averaged across left-out marginals(↓ better) for 5-dim PCA representation of the CITE dataset. Results are averaged across 3 independent runs.

| Method | W1 distance(↓) |
|--------|----------------|
| CFM | $0.538 \pm 0.012$ |
| MFM | $0.571 \pm 0.021$ |
| EGGFM | $\mathbf{0.531 \pm 0.010}$ |

## 6 CONCLUSION

This work takes a step toward making flow-based generative models geometry-aware, by showing how score matching and energy distillation can be harnessed to recover an adaptive metric tensor that faithfully encodes manifold structure. We have demonstrated the utility of calculating a metric tensor through annealed score and energy weighting, applied to synthetic and genomic datasets. Currently, the main limitation of the work is in application to low-dimensional data, and future work aims to generalize other strategies for using generative methods to infer geometry that are more robust at scale. Additionally, the application to genomic data can be further buoyed by downstream analysis of the learned trajectories, in order to understand how the prior knowledge of the data manifold corroborates biological priors about how gene expression changes during differentiation or other temporal processes.

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

## A   EXPERIMENTAL DETAILS

### A.1   SYNTHETIC EXPERIMENTS

Table 4: Base configs for synthetic experiments. If not specified, all subsequent experiments used these hyperparameters as well

| Hyperparameter | Value |
|---|---|
| Learning rate | 0.0001 |
| Hidden Dim | 512 |
| Number of Layers | 4 |
| Gradient Clipping | 10 |
| Score / Energy batch size | 4196 |
| Geodesic / Flow batch size | 256 |
| Frequencies for sinusoidal embedding of noise / cluster id | 32 |
| EMA decay | 0.999 |
| Annealing steps | 2 |
| Metric Scale ($\lambda$) | 10 |
| Number of noise scales in score matching | 20 |
| Min score matching noise($\sigma_{min}$) | 0.01 |
| Max score matching noise($\sigma_{max}$) | 0.2 |
| Metric constant ($\gamma$) | 0.2 |
| Weight beta ($\beta_w$) | 0.3 |
| Energy clip quantiles | [0.05, 0.98] |
| Metric clip lower quantile | 0.05 |
| Flow matching noise ($\sigma_{flow}$) | 0.1 |
| Leiden n_neighbors | 10 |
| Leiden resolution | 0.3 |

For all our architectures, we use MLPs to parameterize the distance embedding, geodesic and flow networks, and MLPs with residual connections for the score and energy networks. For energy we use the parameterization proposed in Thornton et al. (2025) and calculate $E(x) = \langle E_\theta(x), x \rangle$. We also use skip connections in the embedding network.

We consider the sphere in 10 and 20 dimensions, sampled uniformly with a total of 40000 points. For the sake of simplicity, we consider 20000 points in each of two timepoints, and train geodesics, distance and flow losses with a product measure coupling in order to learn geodesics over the entire sphere.

We compare against CFM with comparable parameters using the same architectures as well as MFM, which only requires additional parameters for number of clusters $K = 2000$ and regularization $\kappa = 0.5$, as per Kapusniak et al. (2024).

### A.2   EB EXPERIMENTS

For EB experiments, we use the following configuration for our model:

Table 5: Configuration for EB model

| Hyperparameter | Value |
|---|---|
| Epochs(score net) | 500 |
| Epochs(energy net) | 3000 |
| Epochs(embedding net) | 2000 |
| Epochs(flow net) | 2000 |
| Number of layers | 5 |
| Annealing steps | 3 |
| Metric scale ($\lambda$) | 4 |
| Min temperature | 5.0 |
| Max temperature | 10.0 |
| Min score matching noise($\sigma_{min}$) | 0.1 |
| Max score matching noise($\sigma_{max}$) | 0.2 |
| Flow matching noise ($\sigma_{flow}$) | 0.05 |
| Leiden resolution | 0.0 |

For metric flow matching, we used 500 clusters with $\kappa = 0.1$. To recreate MFM's metric tensor, we skip the score and energy network training. We train the embedding and flow networks with identical hyperparameters. For conditional flow matching, the metric tensor is the identity matrix, hence we also skip the score and energy networks. The rest of the model is trained identically. The data is available at https://data.mendeley.com/datasets/hhny5ff7yj/1 and preprocessed according to the same code as Tong et al. (2020).

## A.3 CITE EXPERIMENTS

We use the CITE donor RNA-seq data publicly available at https://data.mendeley.com/datasets/hhny5ff7yj/1. The data is already preprocessed, so we simply map to 5PCs and whiten each PC. As with EB, we compare against MFM on a performant parameters with $K = 2000$ clusters and $\kappa = 1.0$.

Table 6: Configuration for CITE model

| Hyperparameter | Value |
|---|---|
| Epochs(score net) | 500 |
| Epochs(energy net) | 3000 |
| Epochs(embedding net) | 500 |
| Epochs(flow net) | 2000 |
| Number of layers | 4 |
| Annealing steps | 2 |
| Metric scale ($\lambda$) | 1 |
| Min temperature | 1.0 |
| Max temperature | 1.0 |
| Min score matching noise($\sigma_{min}$) | 0.02 |
| Max score matching noise($\sigma_{max}$) | 0.3 |
| Flow matching noise ($\sigma_{flow}$) | 0.2 |
| Energy clip quantiles | [0.05, 0.95] |
| Metric clip lower quantile | 0.05 |
| Metric constant ($\gamma$) | 0.5 |
| Weight beta ($\beta_w$) | 0.2 |

