# OpenReview forum: "Energy Guided Geometric Flow Matching"
_ICLR.cc/2026/Conference — Submitted to ICLR 2026_

### Official Review · Reviewer_7n8G · 2025-10-29

**Soundness:** 1
**Presentation:** 3
**Contribution:** 2
**Rating:** 2
**Confidence:** 5

**Summary:**

This paper proposes **Energy Guided Geometric Flow Matching (EGGFM)** - method for learning generative flows that adhere to the data manifold's geometry. It learns an energy-guided Riemannian metric and uses it to define geometry-aware couplings and geodesic paths for flow matching, this includes an iterative density refinement stage.

The method is evaluated on synthetic sphere data (10D and 20D) and two single-cell RNA datasets (EB, CITE) against CFM/MFM, reporting W1 in low-dimensional (PCA-5) settings.

**Strengths:**

1. **Iterative density refinement** - The paper correctly identifies that data-density imbalances can distort the learned geometry. The use of importance sampling and stratified sampling to mitigate this and approximate a more uniform manifold density is well-motivated.
2. **Geometry aware coupling** - The authors propose a principled alternative to Euclidean OT by learning geodesics and the associated distance metric, enabling geometry-aware couplings. This is a novel contribution.

**Weaknesses:**

1.  **Preliminary and Low-Dimensional Evaluations** - The empirical evaluation feels preliminary. The synthetic experiments are limited to 10 and 20 dimensions, and the real-world single-cell datasets are evaluated in a 5-dimensional PCA space. This low-dimensional setting is insufficient to validate the method's claims, especially when prior works (e.g. [1,2,3]) evaluated on 50D and 100D as well as additional dataset ("MULTI"). The lack of more comprehensive datasets makes the results feel preliminary.
2.  **Lack of Ablation** - The proposed method is a complex, multi-stage pipeline. However, the paper lacks essential ablation studies to disentangle the contributions of its novel components. It is unclear how much of the performance gain comes from the iterative density refinement versus the energy-based metric itself. Similarly, an ablation on the "geometry-aware coupling" component is needed.
3.  **Lacking comparison for the previous Energy-Based Metrics** - The paper would benefit from a deeper discussion and ablations that motivate the specific metric choice. Energy-based metrics in this context are not new; a direct comparison to [4] is warranted.


[1] Tong, Alexander, et al. "Improving and generalizing flow-based generative models with minibatch optimal transport."
[2] Kapusniak, Kacper, et al. "Metric flow matching for smooth interpolations on the data manifold."
[3] Neklyudov, Kirill, et al. "A computational framework for solving wasserstein lagrangian flows."
[4] Béthune et al., "Follow the Energy, Find the Path: Riemannian Metrics from Energy-Based Models"

**Questions:**

1. **Clustering link** - How does MFM’s RBF kernel centres/clustering relate to your cluster-conditioned iterative density refinement? Where do their effects diverge?
2. **Baseline gaps** -  EB/CITE CFM (and MFM) W1 are lower (better) than published results. Is the setting identical (data splits, PCA dim, coupling, metrics)? Why does MFM < CFM here, and what diagnostics were run? Could this point to suboptimal hyperparameter tuning or an implementation issue for the baselines, which would call the reported performance gains of EGGFM into question? Can the authors confirm that the experimental setting is identical to previous work? If not, what changes were made that could explain this performance discrepancies in the baselines?
3. **More Evidence** - Can you report results on the MULTI dataset and in higher PCA dimensions (50D/100D), ensuring parity with CFM/MFM (same preprocessing/splits), and include brief ablations showing the contribution of iterative density refinement and geometry-aware coupling at these dimensions?

---

### Official Review · Reviewer_abwB · 2025-10-31

**Soundness:** 2
**Presentation:** 2
**Contribution:** 2
**Rating:** 2
**Confidence:** 3

**Summary:**

EGGFM learns a data-driven Riemannian metric from score matching and annealed energy distillation, then uses it to train geodesic-aligned flows via distance learning and flow matching, with stratified sampling to handle disconnected components.  Empirically, it reduces geodesic error on synthetic spheres and lowers W1 distances vs. CFM/MFM on single-cell RNA trajectories, though it’s currently limited to relatively low-dimensional settings.

**Strengths:**

1. Learns a data-driven metric from energy/score models, then uses it to train geodesics and flows.
2. Clear pipeline and motivation; the method is straightforward to follow.
3. Improves geodesic error on synthetic manifolds and W1 on single-cell benchmarks.

**Weaknesses:**

1. Novelty vs. prior metric/density-induced approaches feels incremental and under-theorized.
2. Evaluations are mostly low-dimensional; limited evidence of scalability.
3. Baselines are incomplete.
4. No ablations on key hyperparameters.

**Questions:**

1. How did you pick your metric form? What happens if you change its constants or remove clipping?
2. Which stage drives gains (energy, metric, geodesics, distance, flow)? Please show ablations.
3. Does stratified sampling really help? Show on/off results and different numbers/types of clusters.
4. Why W1 in 5 principle components?

---

### Official Review · Reviewer_bBea · 2025-11-01

**Soundness:** 3
**Presentation:** 2
**Contribution:** 2
**Rating:** 4
**Confidence:** 4

**Summary:**

This paper proposes a flow matching method where trajectories of individual particles remain as close as possible to the underlying data manifold.

The method is composed of multiple steps. The first one is to learn the metric tensor associated with the underlying data manifold. To do so, the authors propose to first learn a sequence of diffusion models that approximate an uniform distribution on the data manifold. Given the corresponding energy, they can define the metric tensor.

With that metric tensor, they can then learn geodesics between two arbitrary points x0 and x1. This trajectories can then be used to train a flow matching model.

The authors evaluate their approach on a synthetic dataset (here they only assess the quality of the learnt geodesics) and on 2 single cell datasets (where they evaluate density interpolation), showing the improved performance of their method compared to baselines.

**Strengths:**

- As mentioned in the introduction, the ability to generate realistic trajectories with generative models is of tremendous interest in biology, where one would wish to examine how one cell evolves from one state to another.
- Flow matching with additional geometric constraints (such as remaining close to the data manifold) is not novel but the authors propose a new way to infer the metric tensor.

**Weaknesses:**

- One weakness of this method is that it requires a lot of different steps that may make actual the whole training process impractical. Each step in the process is also a potential source of errors. Training diffusion models correctly is not trivial, so when applied on another dataset, it's not clear how much this method would work out of the box.
- The mathematical exposition could use more rigor, although I understand that carefully and briefly introducing diffusion models and flow matching in Section 2 is challenging. Section 4.4 seems disconnected from what the text describes.
- The experimental section is quite lean. The experimental setup only evaluates the quality of geodesic. In this case, the authors should compare against the plethora of geometric dimensional reduction methods such as diffusion maps, PHATE, and others (see this review paper as reference https://arxiv.org/abs/2503.05321). While these methods are not aimed at learning transport of distributions, they could plugged in place of the energy estimator introduced in this paper. You should also incorporate more synthetic datasets to establish the advantages of your approach.
- Similarly, the experiments on the single cell datasets should incorporate more of the numerous baselines that have been proposed on this task. Authors mention for instance Rohbeck et al., 2025, that models multiple marginals jointly, and could be a good contender for this task.

**Questions:**

- Could the authors expand the list of baselines for both the synthetic and scRNAseq setups (see my comments above) ?
- Could the authors comment on how much hyper-parameter optimization is needed at each step of the procedure ? It seems the number of parameters is expansive - do authors have semi-automatic ways of choosing them based on the statistics of the dataset ?
- first Eq in Section 4.4 - it seems that you don’t enforce constant speed of the segment in Euclidean space ? Then I assume you do flow matching on that embedding (projecting x with f)  ? But it’s not what the next equation suggests. Can you please clarify this section ?

---

### Official Review · Reviewer_9qLe · 2025-11-01

**Soundness:** 2
**Presentation:** 1
**Contribution:** 1
**Rating:** 2
**Confidence:** 4

**Summary:**

This paper proposes a new score matching model, that is "energy guided"  to learn flows that follows the data manifold. Their method is motivated by applications to single cell data, where there are static snapshot measurements between which generative models have been used for interpolation as well as forecasting.

**Strengths:**

-The paper introduces an interesting idea by combining flow matching with annealed energy distillation. The motivation to constrain flows to the data manifold is valuable, and the framework could inspire future research in this direction.

--The stratified sampling proposed is an interesting way to overcome the issues of disconnected components on score matching models.

**Weaknesses:**

Methods and motivation:
First, the advantage of models like conditional flow matching (Tong et al), Shrodinger bridges (Bunne et al.) and Mioflow (Huguet et al.) for cellular trajectory generation is that they do not have to start from noise and go to data, they can directly start with the first timepoint. This is negated in diffusion models where the training has to start from a known distribution.

GAGA (Sun et al.) actually includes a metric-guided score matching generation, but this is to sample from all areas of the cellular manifold evenly rather than trajectory interpolation.

Second, I cannot understand the advantage of not using the gradient of the data density (score) and plugging the energy which they define as the log of the data density instead.  For pure generation I don't think this would make much of a difference.

Comparisons and related work:

New modern methods have been developed that retrieve trajectories over the manifold that have not been benchmarked against. CFM is not a good benchmark for interpolation as it assumes straight line paths. Some that could be compared are Manifold Interpolating Flows (Huguet et al.) or GAGA (Sun et al). They also cite other methods like Trajectorynet (Tong et al.) which they do not compare to.

I would also encourage the authors to do more comparisons against ground truth trajectories. The baseline of spheres is a  very simple geodesic. There exists single cell simulators that can create a more realistic comparable toy data like SERGIO (https://www.sciencedirect.com/science/article/pii/S2405471220302878).

Presentation:
The methods read in a very dijointed manner. The paragraphs are disconnected. The paper presents multiple loss functions sequentially but provides limited interpretation or intuition for each. It would be helpful to clarify how each term contributes to training stability or manifold alignment.

The Setup and Related Work sections are just copy-pasted from the papers, without proper motivation and connection between paragraphs.

**Questions:**

What is the intuition behind the energy guidance? How is it useful for cellular analysis?

---

### Meta-Review · Area_Chair_XjrM · 2025-12-06

**Summary:**

Reviewers cited lack of comparison to prior work, lack of ablations, and lack of higher dimensional experiments in their reviews. All reviewers agreed that this paper has major weaknesses that must be addressed, but were unfortunately not addressed by the authors.

**Reviewer Concerns:**

No rebuttal was submitted. Therefore all reviewer concerns are still outstanding.

**Reviewer Scores:**

The reviewers would not have changed their scores.

---

### Decision · Program_Chairs · 2026-01-26

Reject